# Risk of Cancer in Connective Tissue Diseases in Northeastern Italy over 15 Years

**DOI:** 10.3390/jcm11154272

**Published:** 2022-07-22

**Authors:** Elena Treppo, Federica Toffolutti, Valeria Manfrè, Martina Taborelli, Ginevra De Marchi, Salvatore De Vita, Diego Serraino, Luca Quartuccio

**Affiliations:** 1Division of Rheumatology, Academic Hospital “Santa Maria della Misericordia”, Azienda Sanitaria Universitaria Friuli Centrale (ASUFC), 33100 Udine, Italy; treppo.elena@gmail.com (E.T.); manfre.valeria.31@gmail.com (V.M.); ginevra.demarchi@asufc.sanita.fvg.it (G.D.M.); salvatore.devita@uniud.it (S.D.V.); 2Department of Medicine (DAME), University of Udine, Via Colugna 50, 33100 Udine, Italy; 3Unit of Cancer Epidemiology, Centro di Riferimento Oncologico di Aviano (CRO), Istituto di Ricovero e Cura a Carattere Scientifico (IRCCS), Via F. Gallini 2, 33081 Aviano, Italy; federica.toffolutti@cro.it (F.T.); mtaborelli@cro.it (M.T.); serrainod@cro.it (D.S.); 4Friuli Venezia Giulia Cancer Registry, Centro di Riferimento Oncologico di Aviano (CRO), Istituto di Ricovero e Cura a Carattere Scientifico (IRCCS), Via F. Gallini 2, 33081 Aviano, Italy

**Keywords:** connective tissue diseases, cancer, lymphoma, Sjogren’s syndrome

## Abstract

Objective: To evaluate cancer risk among individuals with connective tissue disease (CTD) in Friuli Venezia Giulia, northern Italy. Methods: A population-based cohort study was conducted based on data from health records available in the regional healthcare database. Demographic characteristics, hospital discharges, exemption from medical charges, drug prescriptions, were individually matched with data from the population-based cancer registry. Cancer risk was assessed in people diagnosed with the following diseases: systemic lupus erythematosus (SLE), Sjögren’s syndrome (SS), systemic sclerosis (SSc), polymyositis (PM), and dermatomyositis (DM). Results: In all, 2504 patients were followed for a total of 18,006 person-years (median follow-up: 6.8 years). After 5 and 10 years of follow-up, the cumulative cancer incidence was 2.6% and 8.5%, respectively. The most common cancers were breast (*n* = 34), lung (*n* = 24), colon–rectum–anus (*n* = 20), and non-Hodgkin lymphomas (NHL) (*n* = 20). Overall, no excess cancer risk was noted (SIR = 0.87), whereas the number of observed NHL cases was more than two-fold significantly higher than expected (SIR = 2.52). The subgroup analysis showed a higher risk of NHL among SS patients (SIR = 3.84) and SLE patients (SIR = 2.69). Conversely, the study population showed a decreased risk for breast cancers (SIR = 0.61) and corpus uteri (SIR = 0.21). Conclusions: The incidence of NHL was higher among patients with SS and SLE. Careful surveillance for hematological malignancies in these patients is recommended.

## 1. Introduction

Connective tissue diseases (CTDs) include systemic lupus erythematosus (SLE), Sjögren’s syndrome (SS), systemic sclerosis (SSc), polymyositis (PM), and dermatomyositis (DM). Each of them has a typical clinical presentation. According to the current definition, SS is a systemic rheumatic disease, characterized in particular by autoimmune inflammation of the lacrimal and salivary glands, resulting in impaired tear and saliva production. SLE is a systemic autoimmune disease characterized by the production of autoantibodies directed against nuclear and cytoplasmic antigens, which affects several organs and whose clinical presentation is very heterogeneous, ranging from mucocutaneous, hematological and articular symptoms and signs to visceral organ involvement, including the kidney, heart, lung and central nervous system. SSc, also called scleroderma, is an immune-mediated rheumatic disease characterized by fibrosis of the skin and the internal organs and vasculopathy.

Idiopathic inflammatory myopathies (IIM) are systemic inflammatory disorders, which mainly affect the muscles, where muscle weakness, with evidence of muscle damage by increased serum levels of creatine phosphokinase, is usually the classic clinical manifestation, but other organs may be affected, including skin, joints, lungs, heart and gastrointestinal tract.

CTDs are characterized by immune system dysfunction leading to loss of tolerance to self-antigens [1]. Common genetic factors, environmental factors, medical treatment of autoimmune diseases, and impaired immune function have led to speculation of increased cancer incidence in patients with autoimmune diseases [2].

The heterogeneity of both autoimmune diseases and malignancies leads to a complex and bi-directional relationship. On the one hand, an increased risk of malignancies has been demonstrated in some autoimmune diseases [3], and on the other hand, the risk of developing an autoimmune disease may be increased in some malignancies [4]. Among rheumatic autoimmune diseases, this association has been demonstrated particularly in SS, the CTD with the highest risk of lymphoma [5], as well as in other rheumatic diseases [6]. Increased disease activity has been associated with a higher risk of lymphoma in RA [7], whereas such an association is less clear in SLE [8]. Moreover, a biphasic cancer incidence has been observed in SSc, with a first peak occurring early in the disease, suggesting a paraneoplastic phenomenon in which mutant cells serve as autoantigens. A second peak occurs 6 to 8 years after disease onset [9]. Finally, idiopathic inflammatory myopathies (IIMs) are recognized CTDs that have an increased risk of malignancy, justifying the well-accepted definition of cancer-associated myositis (CAM) in the IIM classification [10]. The risk of cancer is especially increased in the 3 to 5 years before and after myositis diagnosis [11], with DM tending to have a higher risk than PM [12]. On the other hand, it is likely that immunosuppressive therapy, which is commonly used in autoimmune diseases, also plays a role in increasing cancer risk [11,13,14].

The present study aimed to assess the risk of cancer associated with definite CTDs in the Friuli Venezia Giulia region of northern Italy during 2002–2017. The primary objective was to determine whether the risk of malignancy was higher in these rheumatic diseases than in the age- and sex-matched general population.

## 2. Materials and Methods

### 2.1. Study Population

A retrospective population-based cohort study was conducted using data from healthcare databases in the Friuli Venezia Giulia region, northeastern Italy (1,206,000 inhabitants). These health-related databases cover the entire regional population and provide de-identified information on demographic characteristics (e.g., sex, age, place of residence), hospital discharges, medical fee waivers, and drug prescriptions. Similarly, data came from the Friuli Venezia Giulia Regional Cancer Registry, which contains data on all new cancer cases from 1995 to 2017. These de-identified databases were linked at the individual level by an anonymous unique key identifier for each individual. Therefore, no ethical approval is required for this type of study.

The cohort included individuals residing in the Friuli Venezia Giulia region who were diagnosed with at least one of the following diseases: SLE, SSc, SS, DM, and PM. To ensure maximum homogeneity and comparability of the exemption codes, the analysis was limited to the years from 2002 to 2017. Patients were identified as CTD cases when they met at least one of the following conditions: (1) a hospital discharge with a CTD diagnosis (codes are listed in the Appendix A); and (2) a medical fee waiver because of a CTD. The date of CTD diagnosis was defined as the first date when one of these conditions was met.

The exclusion criteria were: (1) a follow-up fewer than 90 days; (2) concurrent diagnoses of RA, psoriatic arthritis, or ankylosing spondylitis; (3) had ever used specific medications to treat RA (except rituximab), psoriatic arthritis, or ankylosing spondylitis, i.e., abatacept, adalimumab, anakinra, apremilast, certolizumab pegol, etanercept, golimumab, infliximab, ixekizumab, sarilumab, secukinumab, tocilizumab, and ustekinumab. The Anatomic Therapeutic Chemical (ATC) classification system codes reported in the hospital discharges, the exemption code and the codes reported in the drug orders are shown in Appendix A.

According to these criteria, the study population consisted of 2504 subjects.

Patients were followed from 90 days after the first date on which the diagnosis was mentioned in hospital discharges or leaves of absence and were followed until cancer diagnosis, death, change of regional residence or 31 December 2017, whichever occurred first.

Cancer diagnoses were recorded by the Friuli-Venezia Giulia Cancer Registry according to the rules of the International Association of Cancer Registries and coded according to the International Classification of Diseases, 10th revision (ICD-10). For this study, we excluded NMSC (ICD-10: C44) and diagnoses based only on autopsy or death certificates from the analysis.

### 2.2. Statistical Analysis

Cohort demographic and clinical characteristics were presented as frequencies, medians, and interquartile ranges.

The cumulative cancer incidence function was estimated for the entire cohort and SS patients, treating the death as a competing risk.

To compare cancer incidence in the cohort with the general population, standardized incidence ratios (SIRs) were calculated as the ratio between the observed and expected number of cancer cases. The expected number of cases was calculated using age- and sex-specific incidence rates in the Friuli Venezia Giulia population; 95% confidence intervals (95% CIs) were calculated assuming a Poisson distribution. Analysis was stratified by autoimmune disease and sex; PM and DM were grouped; estimates for patients diagnosed with more than one disease were calculated separately (data not shown). SIRs were reported for cancer sites/types with at least 2 observed cases.

Statistical analyses were performed using SAS Enterprise Guide (version 7.15, SAS, Cary, NC, USA).

## 3. Results

The 2504 individuals (median age of 57 years, IQR: 43–69; 84.2% female) were followed for a total of 18,006 person-years of observation with a median follow-up time of 6.8 years (Table 1). SLE was the most diagnosed disease (documented in 34.6% of study subjects), followed by SS (31.6%), SSc (18.2%), DM, and PM (7.6%). Patients diagnosed with multiple diseases represented 8.0% of the study population.

Figure 1 shows the cumulative cancer incidence functions from CTD diagnosis to 10 years after CTD diagnosis for all patients (*n* = 2504). Cumulative cancer incidence increased steadily with time since diagnosis. After 5 and 10 years of follow-up, i.e., after 5 and 10 years from diagnosis of disease, the cumulative cancer incidence was 2.6% and 8.5% for all patients.

Table 2 lists the SIRs for specific cancer sites/types with at least two observed cases among all patients.

During the study period, 187 cancer cases, other than NMSC, were documented among the cohort, showing no higher risk as compared to the general population (SIR = 0.87, 95% CI: 0.75–1.00). The most common cancers were breast (*n* = 34), lung (*n* = 24), colon–rectum–anus (*n* = 20), and non-Hodgkin lymphoma (NHL) (*n* = 20). When considering NHL, the number of observed cancer cases was more than two-fold significantly higher than expected (SIR = 2.52, 95% CI: 1.54–3.89). It was higher for both males (SIR = 4.54, 95% CI: 1.67–9.89) and females (SIR = 2.12, 95% CI: 1.16–3.55) compared to the corresponding population (data not shown).

The subgroup analysis (Table 3) showed a higher risk of NHL among SS patients (SIR = 3.84, 95% CI: 1.92–6.87) and SLE patients (SIR = 2.69, 95% CI: 0.99–5.84). The incidence of NHL among patients with SS increased from the sixth year after diagnosis (Appendix A).

Conversely, the study population showed a decreased risk for breast cancers (SIR = 0.61, 95% CI: 0.42–0.85) and corpus uteri (SIR = 0.21, 95% CI: 0.03–0.77). Among cancer sites with less than 2 observed cases, a lower risk for ovary cancer (SIR = 0.00, 95% CI: 0.00–0.60) was documented (data not shown).

## 4. Discussion

This population-based study reported the risk of malignancies in patients affected by the most recognized CTDs, specifically SLE, SS, SSc, PM, and DM, over 15 years in northeastern Italy.

Overall, the study showed three main findings: (1) a significantly higher risk of NHL in CTDs, especially in patients affected by SS; (2) no increased risk of site-specific solid cancers in CTDs; and (3) evidence of significantly lower risk of some site-specific solid cancers (i.e., breast, corpus uteri, and ovarian) in CTDs.

Regarding the first point, our results confirmed the previously described association between NHL and CTDs. In our cohort, the incidence of NHL was 2.5 times higher in patients with CTDs than in the general population. This risk was even more significant in SS patients, where the incidence of NHL was almost 4 times higher than in the general population. We did not analyze the main B-cell NHL subtypes. However, it is well-known in SS that lymphoma of mucosa-associated lymphoid tissue (MALT) is the most common histotype of NHL and occurs mainly in the major salivary glands, which are the target tissue of the disease [15,16,17]. In a previous large pooled analysis that included 29,423 subjects, Smedby et al. [3] reported a 6.6-fold increased risk of NHL in SS (which was included in our confidence interval). In addition, the risk of cancer is similar in males and females in our cohort, but slightly higher in males, as previously reported for NHL [18] and especially SS [19]. In our cohort, an increased risk of NHL might also have been assumed in SLE, despite the borderline statistically significant value (SIR 2.69 (95% CI 0.99–5.84)). These results are very close to the large cohort study cited above, which reported a 2.7-fold increased risk of NHL in SLE [3]. The NHL risk was highest in SLE patients with shorter disease duration (2–5 years), whereas the risk was increased almost 2-fold in SLE patients with longer disease duration (≥10 years). Although we were unable to perform an analysis of the main B-cell NHL subtypes in SLE, an increased risk for marginal zone lymphoma (OR 7.5 (95% CI 3.39–16.7)), predominantly of the MALT type, and diffuse large B-cell lymphoma (OR 2.7 (95% CI 1.47–5.11)) in SLE has already been documented in the literature [3]. Overall, our data confirm the known association between the risk of lymphoproliferative processes and autoimmune diseases, especially in SS and SLE, in the Friuli Venezia Giulia population. Since the estimated risk in our population for SS and SLE is very similar to that reported by the InterLymph Consortium, it could be argued that the risk is related to the disease itself and no local environmental factors can influence it.

An increased risk of developing all cancers (except nonmelanoma skin cancer (NMSC)) (SIR 1.32 (95% CI 0.84–1.9)]) was also found in IIMs, although this was not significant. This was probably due to the heterogeneity of the disease. Cancer risk factors should be specifically evaluated in patients with IIM for risk stratification [10]. Myositis autoantibodies are highly specific biomarkers that have prognostic significance and may help in cancer screening. The risk of developing cancer is higher in IIM patients who have anti-TIF1gamma, anti-NXP2, anti-HMGCR or negative autoantibodies [10]. Additionally, DM, increasing age, male gender, dysphagia, and cutaneous ulceration are associated with an increased risk of cancer [10]. Conversely, antisynthetase syndrome appears to be protective.

Regarding the second point (i.e., site-specific solid cancers overall), there was no evidence of a statistically significant increased risk with CTDs. Thus, one could argue that chronic immunosuppressive therapy does not appear to strongly affect the risk of malignancies over time. Nevertheless, a careful preventive approach should be recommended in CTDs. Indeed, it is reasonable to assume that patients with chronic diseases might be induced to pay more attention to national screening programs because of their frequent physician contacts, thereby eliminating potential risk factors, especially for some solid cancers. Screening programs aim to detect early-stage lesions that can be treated before they develop into advanced stages. Therefore, access to screening programs in different countries may also affect cancer registry results. Cancer registries often do not record noninvasive malignancies; one systematic review found a 9-fold increased risk of high-grade squamous intraepithelial lesions of the cervix (HSIL) in SLE women compared with the general population [20], but this does not necessarily translate into an increased risk of cervical cancer in SLE women due to effective screening programs [21]. In our study, SLE patients were not found to be at increased risk for cervical cancer, confirming the overall effectiveness of cervical cancer screening programs in this setting.

Despite the lack of a statistically significant value accounting for all CTDs, our data suggest an increased risk of lung (SIR 1.27 (95% CI 0.82–1.90)) and pancreatic cancer (SIR 1.57 (95% CI 0.83–2.68)), especially the latter with IIM. As for the other autoimmune diseases, i.e., SS, SLE, and SSc, the risk of solid malignancies has only been investigated in a few studies, except for SSc, where the association with cancer, especially lung cancer, was suggested as early as the 1950s [22]. SSc patients have an increased risk of cancer, with SIRs ranging from 1.24 to 4.20 [11]. A recent large cohort analysis [23] found a strong association between older age at SSc onset (OR 1.04 (95% CI 1.02–1.5)), white race (OR 2.71 (95% CI 1.22–6.04)) and overall cancer risk. While the presence of anti-RNA polymerase III antibody was an independent marker of concurrent cancer incidence at any age [23]. In addition to lung cancer, a higher incidence of breast cancer and melanoma was also reported within 5 years of SSc occurrence [24]. A higher risk of esophageal cancer and cancer of the vagina and vulva was also observed [25]. Therefore, careful cancer surveillance should also be recommended in SSc patients, both in the first year of disease and in subsequent follow-up, especially if there has been exposure to cyclophosphamide [26].

According to our third finding, the lower risk for some site-specific solid cancers (i.e., breast, corpus uteri, and ovarian) is of particular interest. The SIRs for breast, corpus uteri, and ovarian cancers were 0.61 (95% CI 0.42–0.85), 0.21 (95% CI 0.03–0.77) and 0.00 (95% CI 0.00–0.60), respectively. Therefore, our results show a lower risk of these malignancies in patients with CTDs compared with the general population, and indirectly support the effectiveness of cancer screening programs in these patients. In addition, some previous studies have supported a lower risk of hormone receptor-negative breast cancer in SLE and also in RA [27].

The strengths of this study include the long data collection period and the possibility of including some rare or even uncommon autoimmune diseases and cancers in a given geographical area. The limitations include the lack of detailed information on disease duration, severity, serological status and medical treatments of CTDs. It is likely that our analysis includes mild disease, and thus underestimated cancer incidence. Medical exemption even for mild diseases may also explain the relatively high number of SLE and SSc patients. As no treatment data were available, we were unable to disentangle the effects of disease and treatment. Unfortunately, we were also unable to adjust for several potential confounders such as smoking, obesity and alcohol consumption.

## 5. Conclusions

This study provides further evidence of a strong association between some CTDs and NHL. The incidence of NHL was significantly higher in patients with primary SS compared to the general population in Italy. Therefore, careful surveillance for hematological malignancies in these patients is recommended.

Overall, our study indicates that physicians should improve cancer screening strategies in CTDs. Finally, investigating the mechanistic basis for the differences in the risk of cancer in patients with systemic autoimmune diseases may improve our understanding of autoimmunity and cancer defense.

## Figures and Tables

**Figure 1 jcm-11-04272-f001:**
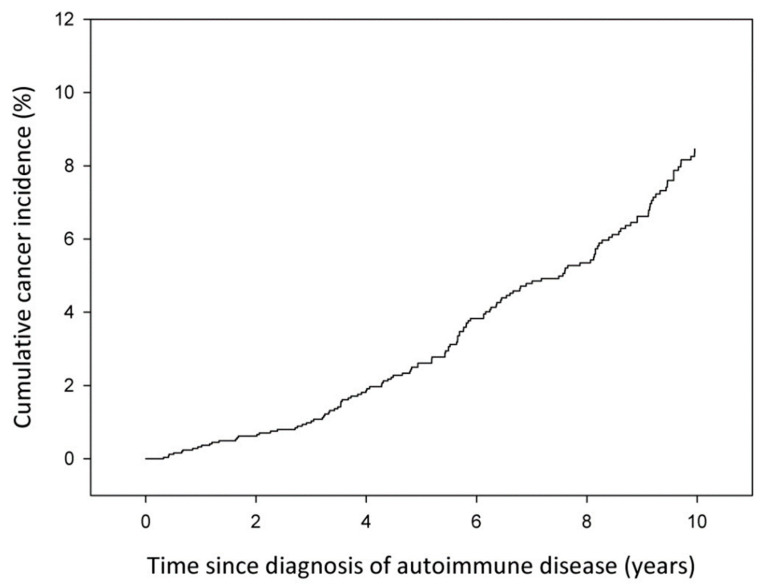
Cumulative cancer incidence by time since diagnosis of autoimmune disease. Friuli Venezia Giulia, 2002–2017. Cancers diagnosed within the first 90 days were excluded.

**Table 1 jcm-11-04272-t001:** Distribution of 2504 patients by selected characteristics. Friuli Venezia Giulia, 2002–2017.

Characteristics	All Patients *n* (%)
Sex:	
Men	396 (15.8)
Women	2108 (84.2)
Age at diagnosis:	
0–29	198 (7.9)
30–39	314 (12.5)
40–49	414 (16.5)
50–59	489 (19.5)
60–69	522 (20.8)
70–79	410 (16.4)
80+	157 (6.2)
Median (IQR *)	57 (43–69)
Period of diagnosis:	
2002–2005	640 (25.6)
2006–2009	676 (27.0)
2010–2013	644 (25.7)
2014–2017	544 (21.7)
Type of autoimmune condition:	
Systemic Lupus Erythematosus	866 (34.6)
Systemic sclerosis	455 (18.2)
Sjogren’s syndrome	791 (31.6)
Dermatomyositis or polymyositis	191 (7.6)
Multiple diagnoses	201 (8.0)
Median follow-up time (years) (IQR *)	6.8 (3.3–10.8)
Total person-years	18,006

* IQR: Interquartile Range.

**Table 2 jcm-11-04272-t002:** Standardized incidence ratios (SIRs) and 95% confidence intervals (CIs) for de novo malignancies by cancer site/type. Friuli Venezia Giulia, 2002–2017.

Cancer Site/Type	All Patients	
	obs./exp	SIR (95% CI)
All, excluding non-melanoma skin cancers ^1^	187/215.4	0.87 (0.75–1.00)
Lip, oral cavity and pharynx (C00–C14)	6/4.6	1.31 (0.48–2.85)
Esophagus (C15)	3/1.8	1.63 (0.34–4.75)
Stomach (C16)	7/9.2	0.76 (0.31–1.57)
Small intestine (C17)	2/0.7	2.81 (0.34–10.2)
Colon, rectum and anus (C18–C21)	20/28.2	0.71 (0.43–1.09)
Liver and gallbladder (C22–C23)	5/6.7	0.75 (0.24–1.75)
Pancreas (C25)	13/8.3	1.57 (0.83–2.68)
Bronchus and lung (C34)	24/18.8	1.27 (0.82–1.90)
Melanoma (C43)	7/7.2	0.98 (0.39–2.01)
Breast (C50)	34/56.0	0.61 (0.42–0.85)
Corpus uteri (C54–C55)	2/9.4	0.21 (0.03–0.77)
Prostate (C61)	4/9.9	0.41 (0.11–1.04)
Kidney and urinary tract (C64–C66, C68)	5/6.5	0.76 (0.25–1.78)
Bladder (C67, D09.9, D41.4)	10/9.7	1.03 (0.49–1.90)
Brain (C71)	2/2.8	0.73 (0.09–2.62)
Thyroid (C73)	7/4.9	1.43 (0.57–2.95)
Non-Hodgkin lymphoma (C82–C85, C96)	20/7.9	2.52 (1.54–3.89)
Multiple myeloma (C90)	5/3.1	1.61 (0.52–3.75)
Leukemia (C91–C95)	3/4.3	0.70 (0.14–2.04)

Obs.: No. of observed cases; Exp.: No. of expected cases. SIRs were presented for cancer site/type with at least 2 observed cases. ^1^ They included cancer cases of thymus (*n* = 1), other connective and soft tissue (*n* = 1), vulva (*n* = 1), cervix uteri (*n* = 1), malignant neoplasm without specification of the site (*n* = 4).

**Table 3 jcm-11-04272-t003:** Standardized incidence ratios (SIRs) and 95% confidence intervals (CIs) for de novo malignancies by cancer site/type and autoimmune condition. Friuli Venezia Giulia, 2002–2017.

Cancer Site/Type	Type of Autoimmune Condition							
	Systemic Lupus Erythematosus		Systemic Sclerosis		Sjögren’s Syndrome		Dermatomyositis or Polymyositis	
	obs./exp	SIR (95% CI)	obs./exp	SIR (95% CI)	obs./exp	SIR (95% CI)	obs./exp	SIR (95% CI)
All, excluding non-melanoma skin cancers	54/62.3	0.87 (0.65–1.13)	35/42.2	0.83 (0.58–1.15)	64/75.5	0.85 (0.65–1.08)	23/17.4	1.32 (0.84–1.98)
Lip, oral cavity and pharynx (C00–C14)	3/1.4	2.13 (0.44–6.22)	2/0.9	2.25 (0.27–8.12)	1/1.5	0.68 (0.02–3.76)	0/0.4	0.00 (0.00–6.94)
Stomach (C16)	3/2.5	1.22 (0.25–3.55)	2/1.9	1.06 (0.13–3.84)	2/3.2	0.62 (0.07–2.23)	0/0.9	0.00 (0.00–3.48)
Colon rectum and anus (C18–C21)	3/7.7	0.39 (0.08–1.14)	6/5.7	1.05 (0.39–2.29)	5/10.1	0.49 (0.16–1.15)	4/2.5	1.63 (0.44–4.17)
Liver and gallbladder (C22–C23)	1/1.9	0.54 (0.01–3.01)	0/1.4	0.00 (0.00–2.16)	2/2.2	0.89 (0.11–3.23)	2/0.7	3.02 (0.37–10.9)
Pancreas (C25)	3/2.1	1.42 (0.29–4.16)	3/1.7	1.75 (0.36–5.11)	4/3.1	1.29 (0.35–3.31)	3/0.7	4.16 (0.86–12.2)
Bronchus and lung (C34)	7/5.4	1.29 (0.52–2.67)	7/3.9	1.82 (0.73–3.74)	6/6.3	0.96 (0.35–2.09)	1/1.9	0.54 (0.01–3.01)
Melanoma (C43)	2/2.4	0.84 (0.10–3.03)	1/1.3	0.79 (0.02–4.40)	3/2.4	1.26 (0.26–3.68)	1/0.5	1.97 (0.05–11.0)
Breast (C50)	11/16.1	0.68 (0.34–1.23)	7/10.2	0.69 (0.28–1.42)	14/21.7	0.65 (0.35–1.08)	1/3.2	0.32 (0.01–1.77)
Prostate (C61)	3/3.7	0.80 (0.17–2.35)	0/2.1	0.00 (0.00–1.41)	1/1.7	0.58 (0.01–3.24)	0/1.6	0.00 (0.00–1.90)
Kidney and urinary tract (C64–C66, C68)	1/1.9	0.53 (0.01–2.97)	2/1.3	1.53 (0.19–5.54)	0/2.3	0.00 (0.00–1.32)	2/0.6	3.42 (0.41–12.4)
Bladder (C67, D09.9, D41.4)	3/2.9	1.05 (0.22–3.06)	2/2.0	1.00 (0.12–3.62)	1/3.0	0.33 (0.01–1.83)	3/1.0	2.88 (0.59–8.43)
Brain (C71)	2/0.8	2.49 (0.30–8.98)	0/0.5	0.00 (0.00–5.61)	0/1.0	0.00 (0.00–3.10)	0/0.2	0.00 (0.00–13.3)
Thyroid (C73)	1/1.8	0.57 (0.01–3.16)	1/0.8	1.29 (0.03–7.18)	5/1.7	3.02 (0.98–7.06)	0/0.2	0.00 (0.00–12.3)
Non-Hodgkin lymphoma (C82–C85, C96)	6/2.2	2.69 (0.99–5.84)	0/1.6	0.00 (0.00–1.92)	11/2.9	3.84 (1.92–6.87)	1/0.6	1.58 (0.04–8.82)
Multiple myeloma (C90)	0/0.8	0.00 (0.00–3.69)	1/0.6	1.57 (0.04–8.76)	3/1.1	2.61 (0.54–7.64)	1/0.3	3.77 (0.10–21.0)
Leukemia (C91–C95)	1/1.2	0.83 (0.02–4.61)	0/0.9	0.00 (0.00–3.51)	2/1.5	1.33 (0.16–4.79)	0/0.4	0.00 (0.00–7.90)

Obs.: No. of observed cases; Exp.: No. of expected cases. SIRs were presented for cancer site/type with at least 2 observed cases.

## Data Availability

Not applicable.

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
