# Peer review of "Risk of Cancer in Connective Tissue Diseases in Northeastern Italy over 15 Years"

_jcm, 2022, doi:10.3390/jcm11154272_

Round 1

Reviewer 1 Report

In this manuscript, the authors assess the incidence of various malignancy in patients with connective tissue disease. The study is well planned and the data is laid out well as well. It might be useful to have the general population level incidence of the various malignancies for comparison in the data table.  It will also be useful to have general population cancer incidence in Figure 1 for comparison. Since the incidence of malignancy increases with age, the authors should comment on how that plays a role in their results. It will be interesting to  ascertain if medications that are used to treat these connective tissue disorders increasing the risk of malignancy.  Overall it is an interesting study.

Author Response

Dear reviewer, thank you for your assessment.

The cohort and the general population have very different distributions by age and sex. To compare cancer incidence in our cohort with the general population, we have considered the standardized incidence ratios (SIRs). SIRs were calculated as the ratio between the observed and expected number of cancer cases. The expected number of cases was calculated using age- and sex-specific incidence rates in the Friuli Venezia Giulia population, and 95% confidence intervals were calculated assuming a Poisson distribution. Therefore, cancer incidence can be compared with general population even if we have not added other data in the table. Figure 1 shows the cumulative cancer incidence since autoimmune disease diagnosis; for the general population there is no date since when we can estimate the cumulative incidence function.

The median age at diagnosis of the selected subjects was 57 years (IQR 43-69), with an almost similar distribution in the different age groups (Table 1). Although the incidence of malignancy increases with age, the data are even more valid because they were compared with the incidence of malignancy in the general population using age- and sex-specific incidence rates. 

Regarding your third comment, we agree with you, further studies could be conduct to ascertain if medications that are used in our cohort to treat these diseases increasing the risk of malignancy. 

Reviewer 2 Report

The study is well designed. However, some minor comments should be addressed.

1. In the introduction section, a brief explanation of each of the connective tissue diseases should be included. 

2. How people enter the study should be clearly explained.

3. Patient follow-up in 5 and 10 years has not been clearly explained.

Author Response

Dear reviewer, thank you for your valuable suggestions.

  1. We have added a brief explanation of each connective tissue disease considered.
  2. We have matched healthcare databases and cancer registry, both referring to the same Italian region (Friuli Venezia Giulia). Therefore, the cohort included individuals residing in the above-mentioned region and affected by systemic lupus erythematosus, Sjögren’s syndrome, systemic sclerosis, polymyositis, or dermatomyositis. Patients were identified as CTDs cases when they met at least one of the following conditions: first, a hospital discharge with an CTDs diagnosis (codes are listed in the supplementary Table 1); second, a medical fee waiver because of CTDs (codes are listed in the supplementary Table 1). The date of CTDs diagnosis was defined as the first date when one of these conditions was met.
  3. This population-based study reported the risk of malignancies in patients affected by the most recognized CTD over 15 years (from 2002 to 2017). The 2504 individuals were followed for a total of 18,006 person-years of observation with a median follow-up time of 6.8 years. We have shown the cumulative incidence function from CTD diagnosis to 10 years after CTD diagnosis (Figure 1) and we have reported in the text the cumulative cancer incidence estimates after 5 and 10 years from diagnosis. We have made this clearer in the text.